# Immunomodulating Enzymes from *Streptococcus pyogenes*—In Pathogenesis, as Biotechnological Tools, and as Biological Drugs

**DOI:** 10.3390/microorganisms12010200

**Published:** 2024-01-18

**Authors:** Lotta Happonen, Mattias Collin

**Affiliations:** Faculty of Medicine, Department of Clinical Sciences, Division of Infection Medicine, Lund University, SE-22184 Lund, Sweden

**Keywords:** *Streptococcus pyogenes*, protease, immunoglobulin (IgG), IgG protease, glycan hydrolase, nuclease, DNAse, immune evasion, immunomodulation, chemokine, complement system, autoimmunity, structural enzymology, biotechnological tool, biological drug, vaccine

## Abstract

*Streptococcus pyogenes*, or Group A *Streptococcus*, is an exclusively human pathogen that causes a wide variety of diseases ranging from mild throat and skin infections to severe invasive disease. The pathogenesis of *S. pyogenes* infection has been extensively studied, but the pathophysiology, especially of the more severe infections, is still somewhat elusive. One key feature of *S. pyogenes* is the expression of secreted, surface-associated, and intracellular enzymes that directly or indirectly affect both the innate and adaptive host immune systems. Undoubtedly, *S. pyogenes* is one of the major bacterial sources for immunomodulating enzymes. Major targets for these enzymes are immunoglobulins that are destroyed or modified through proteolysis or glycan hydrolysis. Furthermore, several enzymes degrade components of the complement system and a group of DNAses degrade host DNA in neutrophil extracellular traps. Additional types of enzymes interfere with cellular inflammatory and innate immunity responses. In this review, we attempt to give a broad overview of the functions of these enzymes and their roles in pathogenesis. For those enzymes where experimentally determined structures exist, the structural aspects of the enzymatic activity are further discussed. Lastly, we also discuss the emerging use of some of the enzymes as biotechnological tools as well as biological drugs and vaccines.

## 1. Introduction

*Streptococcus pyogenes*, or Group A *Streptococcus*, is an exclusively human pathogen that causes a wide variety of diseases ranging from mild throat and skin infections to severe invasive disease. *S. pyogenes* is one of the leading etiological agents in infection-related deaths worldwide [1], and is one of the most common human pathogens, estimated to account for 15–30% of all pharyngitis cases in children and 5–10% in adults [2]. Furthermore, especially in warm and humid climates, the bacterium causes skin and soft tissue infections such as impetigo, erysipelas, and cellulitis [3]. Most infections are relatively mild and self-limiting, but major health concerns are the progression to infections in deeper tissue (necrotizing fasciitis) and systemic spread (sepsis), and autoimmune sequelae affecting the joints and heart (acute rheumatic fever), as well as the kidney (acute post-streptococcal glomerulonephritis) [4]. The pathogenesis of *S. pyogenes* infection has been extensively studied, but the pathophysiology, especially of severe infections, is still somewhat elusive (reviewed in [5,6,7]).

One key feature of being a successful pathogen is the ability to evade or modulate host immunity against the bacterium itself. *S. pyogenes* expresses a multitude of secreted, surface-associated, and intracellular molecules that directly or indirectly affect the immune system (excellently reviewed in [8]). A distinct subgroup of immune evasion factors are enzymes that directly target parts of innate and adaptive immunity. Almost exactly 20 years ago, one of us got the opportunity to give an overview of the immunomodulating enzymes in *S. pyogenes* [9]. Much has happened since, with the identification of novel immunomodulating enzymes, the elucidation of structure/function relationships, and the uncovering of roles during *S. pyogenes* infections. Undoubtedly, *S. pyogenes* present itself as one of the major bacterial sources for such enzymes. We also find it especially intriguing that some of these enzymes have proven to be valuable biotechnological tools (on the market and under development) and biological drugs against autoimmunity (approved and experimental). In this review, we attempt to present current knowledge about immunomodulating enzymes in *S. pyogenes*. It is not possible to go into all the molecular details about all the enzymes, but we are hoping to inspire the readers to further reading. Here, the focus is on enzymes with reasonably direct immunomodulating activities, such as in immunoglobulin (Ig)-degrading and Ig-modifying enzymes, enzymes interfering with the innate immune system, including the complement system, and enzymes acting on chromatin and cellular processes (For a visual summary, see Figure 1). The delineation between strict immunomodulating enzymes and metabolic enzymes is not always clear, since a common feature of bacterial pathogens is the use of common metabolic enzymes for other purposes, including immune evasion. These so-called moonlighting enzymes are outside the scope of this review, since most of their functions in pathogenesis are not related to their enzymatic activities (for a review, see [10]).

## 2. Main Functional Categories of *S. pyogenes* Immunomodulating Enzymes

### 2.1. Immunoglobulin Degrading and Modifying Enzymes

Without adaptive immunity, bacterial infections would have eradicated humanity a long time ago. It is therefore not surprising that antibodies are major targets for bacterial attack. Bacteria showcase a multitude of factors interfering with antibody-mediated effector functions, including enzymatic activities [11]. *S. pyogenes* expresses a remarkable number of proteins targeting and modulating human antibodies and their effector functions. These include immunoglobulin-binding surface proteins such as the M- and M-like proteins (not covered here, but for review, please see [4,11]) and unique enzymes targeting human antibodies, which will be the focus here (Table 1).

#### 2.1.1. Immunoglobulin Cysteine Proteinases

##### The Broad-Spectrum Cysteine Proteinase SpeB

The broad-range cysteine proteinase SpeB is one of the major secreted proteins in *S. pyogenes* (for the history of its discovery and naming, see [12]). It is secreted as a 40 kDa zymogen that is autocatalytically cleaved into the 28 kDa active proteinase (Figure 2A). SpeB has been implicated to have a role in the pathogenesis of *S. pyogenes* infections in animal models and in clinical studies (for reviews and references therein, see [12,13]) (Figure 1, Table 1). SpeB has in vitro activity on a number of human proteins in the extracellular matrix and plasma (for a review see [12]). However, the physiological relevance of some of these activities can be debated.

We have demonstrated that SpeB is capable of breaking down immunoglobulins (Ig) A, M, D, and E into smaller fragments, and that it can also sever IgG in the hinge region to create fragment crystallizable (Fc) and fragment antigen-binding (Fab) domains [14,15]. Additionally, studies have revealed that SpeB specifically digests IgG that is antigen-bound on bacterial surfaces via its Fab-domain, while leaving IgG that is non-immunologically attached via Fc-binding surface proteins intact [16]. The separation of the Fc and Fab fragments in IgG predictably results in a reduction in the antibody-mediated opsonophagocytosis of *S. pyogenes* in vitro [17]. Whether the SpeB hydrolysis of antibodies is physiologically relevant has been challenged (and we tend to agree) [18], but in vitro activity is of biotechnological interest in any instance (see below).

##### The Immunoglobulin G-Degrading Cysteine Proteinases IdeS/Mac-1/Mac-2

The secreted 35 kDa enzyme IdeS/Mac-1 was almost simultaneously identified by two independent research groups. One group focused on the similarities with the human protein Mac-1 (CD11b) and denoted it Mac-1, and described its antiphagocytic properties [19]. The other group showed that the protein is a cysteine proteinase with a remarkable specificity for human IgG, and denoted it immunoglobulin G-degrading enzyme of *S. pyogenes,* or IdeS [20]. An allelic variant of IdeS/Mac-1 denoted Mac-2 is also present in certain serotypes, as well as an enzymatically impaired variant in some serotype M28 strains [21]. However, the IgG protease activity seems to be the dominating feature overall [22]. IdeS has been extensively characterized, and it hydrolyses IgG in the lower hinge under the disulphide bridges generating a F(ab’)_2_ and two ½ Fc fragments (Figure 1) [23,24]. Determination of the crystal structure of IdeS both alone and in complex with IgG-Fc has shed light on its strict specificity for IgG [25,26] (Figure 2B). The strict specificity for IgG has previously been suggested to be mediated by exosite binding and interaction with another region outside of the active site [23]. This has been verified by the structural model of IdeS in complex with IgG, where IdeS has been demonstrated to bind across both IgG chains in the Fc region. With IdeS binding, the Fc domain peptide backbone becomes distorted, promoting cleavage between two glycine residues (Figure 2B) [26].

**Figure 2 microorganisms-12-00200-f002:**
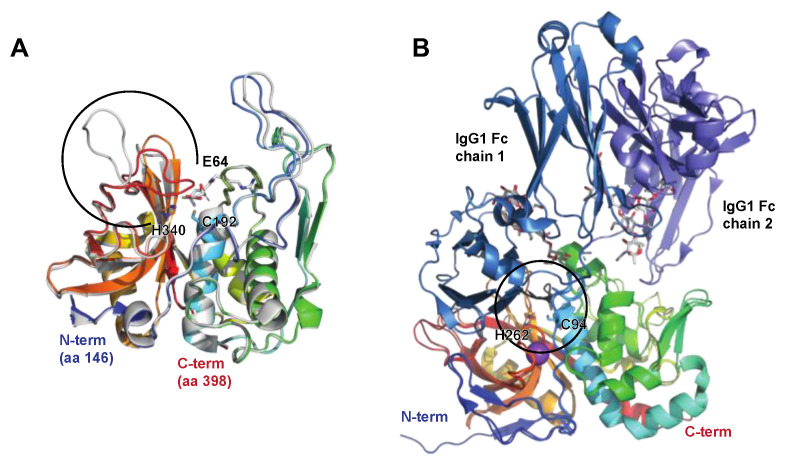
The streptococcal cysteine proteases SpeB and IdeS. SpeB and IdeS are both immunoglobulin cysteine proteinases with a papain fold. Common to such proteases is a catalytic site composed of a Cys and a His residue that exist as a zwitterionic pair [27]. (**A**) Structure of streptopain SpeB, PDB IDs: 4D8B (apo; shown in rainbow) and 4D8E (in complex with the small molecule inhibitor E64, shown in grey [28]). SpeB is composed of an N-terminal signal sequence (aa 1–27), a prodomain (aa 28–145), and a catalytic C-terminal region (aa 146–398), the structure of which is depicted here. SpeB has a unique, highly flexible glycine-rich C-terminal loop (aa 368–390, indicated by a black open circle), which, in the absence of a ligand, is positioned over the active sites Cys192 and His340 [27], and the side chains of which are shown as sticks. For a detailed view of the active site and the conformational changes associated with SpeB activation and substrate active site binding, please see the detailed description in [28]. (**B**) Structure of IdeS (Mac-1 and Mac-2), containing a Cys94 mutation abolishing catalytic activity, alone and in complex with the IgG1 Fc fragment, PDB IDs: 1Y08 (apo [25], shown in rainbow) and 8A47 (with IgG [26], shown in rainbow (IdeS), shades of blue (IgG heavy chains), and grey (heavy chain-attached glycans)). IdeS cleaves IgG molecules at a defined sequence between two Gly residues (shown as black sticks) in the lower hinge region in the heavy chain. This cleavage takes place in two distinct steps, with the cleavage of the first chain being faster. In the crystal structure, IdeS can be seen to clamp down over the lower hinge region of one of these Fc chains, creating a cavity in which the catalytic residues (Cys94Ser [PDB ID: 1Y08] or Cys94Ala [PDB ID: 8A47] and His262, sidechains shown as sticks) are brought into close proximity to the Gly-Gly cleavage site (black circle).

#### 2.1.2. Immunoglobulin Glycan Hydrolases

##### The Endo-β-*N*-acetylglucosaminidase EndoS

When EndoS was discovered, it represented the first described enzyme from *S. pyogenes* with specificity for human IgG and also the first described bacterial antibody glycan hydrolase [14]. The *ndoS* gene is found in the vast majority of genome-sequenced *S. pyogenes* strains and is highly conserved. EndoS is a 108 kDa secreted endoglycosidase belonging to family 18 of glycosyl hydrolases and hydrolyzes the chitobiose core of the conserved *N*-linked glycan on Asn297 in the heavy chain on human IgG. EndoS displays similarity to other endo-β-*N*-acetylglucosaminidases from bacteria, most notably EndoF_2_ from *Elisabethkingia meningoseptica* (formerly *Flavobacterium meningosepticum*), which has been extensively used as a glycan mapping tool [29]. Interestingly, the other members belonging to this family of enzymes are not dependent on the IgG protein backbone for activity, while EndoS only hydrolyzes the *N*-linked glycan if it is attached to Asn297 of the IgG heavy chain. It does not, for instance, hydrolyze *N*-linked glycans if those are present in the variable regions of the heavy and light chains of IgG. EndoS has activity on most naturally occurring IgG heavy chains’ glycans (complex-type, biantennary), but has limited activity on bulkier glycans, such as high-mannose and hybrid-type glycans, including glycans with bisecting GlcNAc [30]. Structural determination of EndoS both alone and in complex with IgG-Fc has uncovered that its extreme specificity for IgG is dependent on both interactions between the catalytic domain and the IgG glycan as well as additional protein–protein interactions between EndoS and the constant domain of human IgG, in analogy with the exosite binding described for IdeS above [26,31,32,33] (Figure 3A,B).

##### The Endo-β-*N*-acetylglucosaminidase EndoS2

EndoS2 is an allelic variant of EndoS that has been identified in serotype M49 of *S. pyogenes* that displays only approximately 40% amino acid identity to EndoS. The *ndoS2* gene replaces the *ndoS* gene in M49 strains and is in the same genetic context as *ndoS* in other serotypes. EndoS2 is also somewhat smaller, with an approximate size of 95 kDa, with the key structural difference to EndoS being the lack of two α-helices located towards the C-terminal end of the protein (Figure 3A,C). EndoS2 has activity on all subclasses of human (and several other subclasses of mammalian) IgG, but has, in addition, activity on the serum glycoprotein α_1_-acid glycoprotein [34]. Interestingly, EndoS2 has a much broader specificity for IgG-Fc glycans than EndoS by hydrolyzing bulkier glycans, such as high-mannose and bisecting structures [30]. Structural determination of EndoS2 in complex with glycan substrates showed a different type of interaction between the glycan and the enzymatic domain compared to EndoS [32], partly explaining the difference in glycan specificity (Figure 3D). Furthermore, a recent preprint presenting the structure of EndoS2 in complex with IgG-Fc further stresses the importance of the non-enzymatic domains for specific activity (PDB not released at the time of writing of this review article) [35].

**Figure 3 microorganisms-12-00200-f003:**
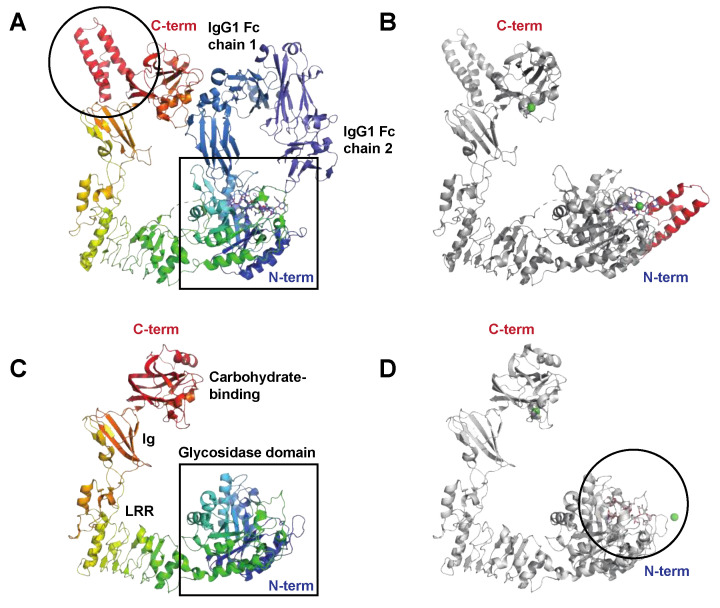
The streptococcal immunoglobulin glycan hydrolases EndoS and EndoS2. (**A**) Structure of the catalytically inactive EndoS Asp233Ala/Glu235Leu mutant in complex with the IgG1 Fc fragment, PDB ID: 8A49 [26], shown in rainbow (EndoS), shades of blue (IgG heavy chains), and blue (heavy chain-attached glycans). EndoS is composed of a proline-rich loop (aa 98–112), a glycosidase domain (aa 113–445), a leucine-rich repeat domain (LRR) (aa 446–631), an Ig domain (aa 632–764), a carbohydrate-binding domain (aa 765–923), and a C-terminal three-helix bundle domain (aa 924–995); the helix bundle is indicated (black circle). The catalytic center resides within the N-terminal glycosidase domain (black square). (**B**) Structure of EndoS in complex with a G2 oligosaccharide, PDB ID: 6EN3 [36] show in grey. In contrast to the EndoS crystallized together with the IgG Fc, which only contained EndoS amino acids 98–995, the structure in complex with the G2 oligosaccharide (blue sticks) also contains additional N-terminal residues that fold into an additional helical bundle, indicated here in red. Green spheres are Ni^2+^ and Ca^2+^ ions. (**C**) Structure of EndoS2, PDB ID: 6E58 [36] shown in rainbow colors. Like EndoS, EndoS2 has a glycosidase domain (aa 43–386) followed by a LRR domain (aa 387–547), an Ig domain (aa 548–680), and a carbohydrate-binding domain (aa 681–843) (all indicated in the figure). The main structural difference compared to EndoS is the lack of the C-terminal three-helix bundle domain, shown in panel (**A**). (**D**) Structure of EndoS2 (shown in grey) in complex with a high-mannose glycan (blue sticks), PDB ID: 6MDV [36]. The high-mannose glycan binds in a cleft (black circle) in the glycosidase domain. For a detailed description of the interaction interfaces between EndoS2 and its substrate glycan(s), please see [32].

### 2.2. Enzymes Interfering with Innate Immunity

Innate immunity is the first line of defense against invading pathogens in the body. It consists of a preformed defense mechanism that is present from birth and provides immediate protection. Innate immunity is mediated by various components, including physical barriers like the skin and mucous membranes as well as cellular and molecular mechanisms.

To combat the effects of the innate immune system, *S. pyogenes* expresses several enzymes degrading both complement components as well as chemokines and cytokines (Table 1). In addition to expressing complement-degrading enzymes, *S. pyogenes* expresses other virulence factors inhibiting complement functions, such as the recently described complement evasion factor (CEF) and streptococcal inhibitor of complement (SIC) [37,38], both outside the scope of this review. Furthermore, proteolysis and modification of extracellular matrix (ECM) components that can alter barrier functions will not be discussed here (for a summary of SpeB’s activities on ECM, see [12]). *S. pyogenes* has also developed means of handling reactive oxygen species that that are generated as metabolic end products and as distinct antibacterial effectors. This is a whole research field in itself that has recently been excellently reviewed in [39] and will not be covered beyond the mention here and listing in Table 1.

#### 2.2.1. Complement and Antimicrobial Peptide Degrading Enzymes

##### The Broad-Spectrum Cysteine Proteinase SpeB

The cysteine proteinase SpeB has been found to act on key components in both the classical and alternative pathways of complement activation. C3b, an effective opsonin that attracts phagocytic cells to infection sites, has been notably absent around soft tissue infections caused by *S. pyogenes*. It has also been observed that C3b levels are reduced in the blood serum of patients with streptococcal toxic shock syndrome (STSS), and that SpeB is capable of breaking down C3b in vitro [40]. It has further been demonstrated that SpeB degradation of C3b hampers both opsonization and the subsequent phagocytic destruction of bacteria [41]. Moreover, SpeB can disrupt the alternative pathway by breaking down properdin, an enhancer of complement activation [42].

##### The Serine Endopeptidase C5a, ScpA

The C5a peptidase, ScpA, from *S. pyogenes* is one of the most extensively researched enzymes that modulate immune response (Figure 1). This enzyme specifically targets a key factor of human immunity: the chemotactic complement factor C5a. ScpA is a 130 kDa serine endopeptidase (Figure 4A) anchored to the cell wall that specifically cleaves C5a [43]. By doing so, it hampers the ability to attract phagocytic cells to the site of infection [44]. C5a is also instrumental in activating neutrophils that ingest the bacteria, highlighting the significance of ScpA’s activity [45]. An intriguing observation is that SpeB can liberate functional ScpA fragments, which then go on to neutralize C5a remotely from the bacterium itself [46].

#### 2.2.2. Chemokine, Cytokine, and Kinin Active Enzymes

##### SpeB Activity on Immunologically Active Peptides

Chemokines are small proteins secreted from a variety of immune and non-immune cells. Not only are they powerful molecules for leukocyte signaling and differentiation, some of them also possess inherent antibacterial properties [47]. We have found that SpeB can destroy most of the signaling and antibacterial capabilities of chemokines released from inflamed epithelial cells, likely contributing to the survival of *S. pyogenes* in the host [48]. While most of the activities of SpeB discussed so far have anti-inflammatory effects, it has additional pro-inflammatory properties. SpeB has the ability to convert pro-interleukin-1β (IL-1β) into its biologically active form, triggering a state of inflammation that could worsen the effects of *S. pyogenes* infection (Figure 1) [49]. Another target of SpeB is H-kininogen, a component of the contact activation system that acts to inhibit proteinases. SpeB’s enzymatic action on H-kininogen, both in vitro and in vivo, leads to the release of bradykinin, a powerful vasodilator and pain inducer, which is likely to play a role in the pathogenesis of invasive *S. pyogenes* infections (Figure 1) [50]. In addition to targeting chemokines, IL-1β, and H-kininogen, SpeB has additionally been observed to induce the release of histamine and cause degranulation in mast cells [51]. While the exact mechanism behind its impact on mast cells remains unclear, it appears to be related to SpeB’s proteinase activity and may contribute to conditions such as STSS.

##### The Serine Proteinase SpyCEP

SpyCEP was originally identified as an enzyme with IL-8-degrading protelytic activity, regulated by a pheromone peptide and found in necrotizing soft-tissue isolates [52]. Subsequently, it was proven to be a cell-wall-anchored serine proteinase, produced as a 170 kDa zymogen, that is autocatalytically cleaved into 150 and 30 kDa forms that together form the active proteinase [53,54] (Figure 4B). SpyCEP is anchored to the cell wall by sortase, but is also released from the surface in the later growth phase [55,56]. Much like SpeB, SpyCEP also targets chemokines. SpyCEP hydrolyzes a group of ELR (Glu-Leu-Arg)-motif-containing CXC chemokines, including the strong neutrophil chemoattractant CXCL8 (IL-8) [57,58,59]. This contributes to *S. pyogenes* suppression of neutrophil migration into the site of infection (Figure 1) and consequently avoidance of opsonophagocytosis [60]. Intriguingly, in addition to hydrolyzing chemokines, SpyCEP also hydrolyzes the antimicrobial and signaling peptide LL-37, leading to reduced signaling activity through host receptors such as the nucleotide receptor P2X7R and the epidermal growth factor receptor (EGFR), while not affecting antibacterial activity [61,62].

**Figure 4 microorganisms-12-00200-f004:**
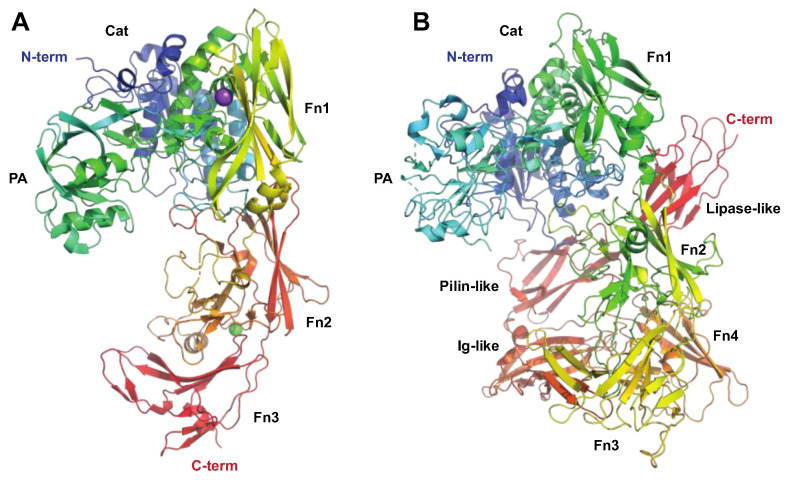
The streptococcal subtilisin-like proteases ScpA and SpyCEP targeting the innate immune system. The C5 peptidase (ScpA) and SpyCEP are two pseudo-paralogous cell-surface-associated subtilisin-like proteases, which both are involved in evasion of the innate immune system. Common to these proteins is that they both target chemokines. (**A**) Structure of C5a peptidase, PDB ID: 3EIF [63] shown in rainbow colors. ScpA has a signal peptide, followed by a subtilisin-like catalytic domain (Cat, aa 32–583, shown in shades of blue and green) with an inserted protease-associated (PA) domain in the middle, and C-terminal A domains with fibronectin (Fn) type III domains (aa 584–1032, shown in shades of yellow to red) and W (aa 1033–1126) domains, the latter excluded in the structure. Even though ScpA is capable of degrading chemokines, its primary target is the complement component C5a. Structural modeling of the complex between ScpA and C5a suggests that the core of C5a is in close proximity to one of the Fn domains, with the C5a C-terminal tail being predicted to extend through the ScpA active site [63]. As for IdeS and IgG, the specificity of ScpA to C5a is likewise suggested to be mediated both by active site specificity and exosite interactions with the Fn domain, with the exosite interactions mediated by electrostatic forces. (**B**) Structure of SpyCEP, PDB ID: 5XYR [62] shown in rainbow colors. Like ScpA, SpyCEP has a signal peptide, followed by a subtilisin-like catalytic domain (Cat, aa 116–420, shown in dark blue) with an inserted protease-associated (PA) domain in the middle (aa 420–580, turquoise and green), followed by four fibronectin (Fn)-like domains (aa 580–1285, green to orange). Unlike ScpA, SpyCEP harbors three additional C-terminal domains (orange to red): a reverse-Ig-like domain (aa 1285–1398), a pilin-like domain (aa 1398–1492), and a lipase-like domain (aa 1492–1574).

### 2.3. Enzymes Acting on Chromatin and Cellular Processes

The enzymes we have covered so far in this review include those acting on proteins (immunoglobulins, chemokines, and complement factors) or glycans. A specific group of *S. pyogenes*-encoded immunomodulatory enzymes, however, target host DNA, nucleotides, or the coenzyme nicotinamide adenine dinucleotide (NAD) (Table 1). *S. pyogenes* encodes for at least eight different nucleases, as relatively recently reviewed [64,65]. Even though these DNAses are abundant among the pathogenic streptococci, the role of these enzymes in streptococcal pathogenesis is in part still unclear. The known functions of these DNAses mainly involve the degradation of host DNA in neutrophil extracellular traps or NETs [66,67]. Here, we discuss the functions of three of these DNAses. The enzymes targeting nucleotides mainly function in converting substrates into the immunomodulators adenosine or deoxyadenosine, the latter capable of inducing apoptosis in macrophages and monocytes. Enzymes degrading NAD or generating potent second messengers are also discussed.

#### 2.3.1. Enzymes Acting on DNA and Nucleotides

##### The Streptococcal Nuclease Sda1

The prophage-encoded streptococcal nuclease Sda1 (Figure 5) was originally described as a secreted DNAse expressed by the M1T1 clone and is homologous to streptodornase D (SdaD) [68,69]. Like other streptococcal DNAses, Sda1 has been shown to promote streptococcal neutrophil resistance via the degradation of host DNA in NETs (Figure 1) in a murine model of necrotizing fasciitis [66]. In addition to degrading host DNA, Sda1 can dampen the host’s immune response by degrading streptococcal CpG-rich DNA. This suppresses Toll-like receptor 9 (TLR9)-mediated IFN-α and TNF-α production, hence decreasing macrophage bactericidal activity [70].

##### The *Streptococcus pyogenes* Nuclease A, SpnA

SpnA has been described as a cell-wall-associated DNAse with functions in the degradation of NETs (Figure 1) [67,72]. SpnA is homologous to the cell-wall-associated DNAse SsnA, the secreted nuclease A of *Streptococcus suis* [72]. Whereas no experimental structure for SpnA exists to date, the N-terminal domain is known to harbor at least three oligonucleotide-binding (OB) domains required for DNA binding during NET degradation, and, based on in vitro assays using recombinant proteins, at least two of these domains (OB2 and 3) are required for SpnA activity [67]. The C-terminal part of SpnA contains the DNA-degrading endo/exonuclease domain [67]. The DNase activity (and structural integrity) of SpnA is dependent on Ca^2+^ and Mg^2+^ ions [72]. Intriguingly, our recent findings demonstrate that SpnA binds the immunoglobulin G-degrading cysteine proteinase IdeS, and that, when in complex with SpnA, IdeS efficiently cleaves IgG antibodies, an interaction promoting IgG clearance at the bacterial surface [73].

##### The Streptococcal 5′-Nucleotidase A, S5nA

S5nA is a cell-wall-anchored ecto-5′-nucleotidase [74] that efficiently converts adenosine monophosphate (AMP) and, to a lesser extent, adenosine diphosphate (ADP), but not adenosine triphosphate (ATP), into the immunomodulator adenosine, facilitating the evasion of bacteria from the host immune response (Figure 1) [74]. Similar nucleotide-degrading enzymes have been described in other Gram-positive bacteria as well, such as the closely related *Streptococcus suis*, Group B *Streptococcus* (NudP), and *Staphylococcus aureus* (AdsA) [75,76]. As has been described for NudP and AdsA, S5nA can additionally convert deoxyadenosine monophosphate (dAMP) into deoxyadenosine (dAdo). dAdo is capable of inducing caspase-3-mediated apoptosis in macrophages and monocytes. Intriguingly, S5nA and SpnA have been demonstrated to generate inorganic phosphate from DNA, suggesting that these two enzymes cause a so-called “double-hit” to the host’s innate immune response by destroying NETs and killing macrophages via generation of dAdo [74], as has been described in *S. sanguinis* [77]. It has, however, further been demonstrated that S5nA does not significantly affect streptococcal growth in human blood, evasion of phagocytosis, or the formation of biofilm [78], suggesting that the main role of S5nA in streptococcal pathogenesis is indeed the destruction of macrophages.

##### The Streptococcal NAD^(+)^-Glycohydrolase, NADase

The *S. pyogenes*-encoded nicotinamide adenine dinucleotide (NAD)–glycohydrolase, also known as NADase, NGA, or SPN, is introduced into the host cell’s cytoplasm via cytolysin-mediated translocation (CMT) using another streptococcal virulence factor, streptolysin O (SLO), as a portal (Figure 1) [79]. NADase has several different activities contributing to streptococcal pathogenesis, including its ability to cleave β-NAD^+^ at the ribose–nicotinamide bond to generate ADP-ribose and the potent vasoactive compound nicotinamide as well as its ability to convert β-NAD^+^ into cyclic ADP-ribose, a potent second messenger. In order to protect itself against the detrimental effects of NADase, *S. pyogenes* expresses an endogenous inhibitor of NADase, IFS [80] (Figure 6). IFS inhibits glycohydrolase activity by acting as a competitive inhibitor of the β-NAD^+^ substrate.

##### The C-di-AMP Synthase DacA

C-di-AMP synthases like DacA, found in many bacteria, are responsible for synthesizing cyclic-di-adenosine monophosphate (c-di-AMP), which is a second messenger molecule involved in various cellular processes. In *S. pyogenes*, it has been shown to have indirect immunomodulatory effects through the regulation of several pathways, including SpeB expression [83]. However, it has recently been shown to be involved in the activation of a type I interferon response mediated through the intracellular sensor stimulator of interferon genes (STING) in macrophages, contributing to disease severity. This is pronounced in individuals with a STING genotype, leading to reduced c-di-AMP binding in combination with *S. pyogenes* strains with high NADase activity [84].

#### 2.3.2. Enzymatic Effects on Pyroptosis and Autophagy

As highlighted throughout this review, the cysteine protease SpeB has multiple roles in streptococcal pathogenesis. We have so far described its role in the degradation of immunoglobulin, complement system components, and chemokines. In addition to this, SpeB is involved in immune modulation by inducing pyroptosis and inhibiting autophagy, as detailed below.

##### SpeB Activity on Gasdermins

Gasdermins are host proteins involved in a process called pyroptosis, which is a type of cell death associated with inflammation and the host’s response to bacterial infections [85]. Gasdermins are cleaved by various proteases, including caspases, to release their N-terminal fragments, which form pores in the cell membrane, leading to cell lysis and inflammation. It has recently been shown that SpeB can activate gasdermins in keratinocytes that undergo pyroptosis, leading to killing of the bacteria, and that mice lacking gasdermins are more susceptible to infection (Figure 1) [86]. This most likely represents an ancient sensing mechanism for dangerous proteases, and represents an immunomodulating activity of SpeB that is beneficial for the host rather than for *S. pyogenes*.

##### SpeB Activity on Ubiquitin-Binding Proteins

Ubiquitin-binding proteins are involved in various cellular processes, including protein degradation, DNA repair, and signal transduction. They recognize and interact with ubiquitin moieties attached to target proteins leading to the controlled destruction, autophagy, of labeled proteins. Autophagy is recognized as a vital component of the innate immune response against intracellular bacteria. It has recently been shown that *S. pyogenes* has the capacity to evade ubiquitylation and evade recognition by the host autophagy marker LC3, as well as the ubiquitin-LC3 adaptor proteins NDP52, p62, and NBR1 [87]. This phenomenon depends upon the expression of SpeB, as an isogenic mutant of M1T1 lacking SpeB was more susceptible to autophagic targeting and demonstrated reduced intracellular replication [87]. In the same study, SpeB was shown to directly degrade p62, NDP52, and NBR1.

#### 2.3.3. Enzymes Acting on Other Cellular Processes

##### The Streptococcal Arginine Deiminase SAGP

Like many other bacteria, *S. pyogenes* generates a diverse array of enzymes from different categories, including metabolic and housekeeping enzymes. Most of these enzymes have no proven direct link to immune evasion. However, among these enzymes, the streptococcal acid glycoprotein (SAGP) stands out (Figure 7). Initially, SAGP was characterized as an antitumor protein [88]. It exhibits arginine deiminase activity (breakdown of arginine to citrulline and NH_3_) and suppresses the proliferation and differentiation of T-lymphocytes in vitro (Figure 1) [89,90]. Furthermore, SAGP has been suggested to play a role in the survival of *S. pyogenes* under acidic conditions though the buffering activity of NH_3_, potentially aiding in intracellular/lysosomal survival [91].

**Figure 7 microorganisms-12-00200-f007:**
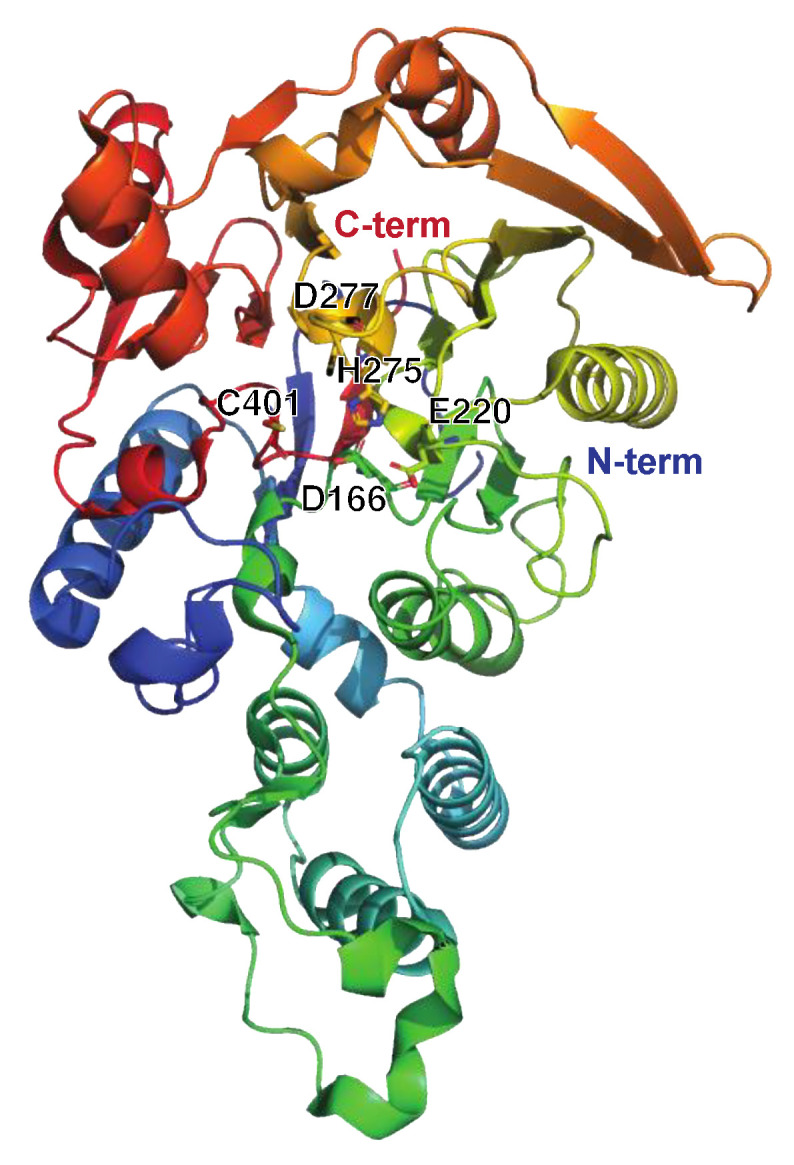
Structure of the streptococcal acid glycoprotein SAGP or arginine deiminase ADI, PDB ID: 4BOF ([92], shown in rainbow colors). The active site residues Asp166, Glu220, His275, Asp77, and Cys401 are indicated. Individual mutation of these residues impaired the catalytic activity of ADI as determined by arginine conversion to citrulline. Intriguingly, the Asp211Ala and Asp277Ala mutants were deemed the most suitable as vaccine candidates due to their preserved structural integrity compared to wt ADI, as well as the production of high-titer antisera against these. Moreover, neither mutant disrupted the B-cell epitopes of ADI. Importantly, there is no human ortholog of ADI, further strengthening its possible use as a future vaccine target.

**Table 1 microorganisms-12-00200-t001:** Immunomodulating enzymatic activities in *S. pyogenes*.

Enzyme	Type of Enzyme, Designations, and Structure Used for Illustrations	Occurrence, Immunomodulating Activities, and Role in Pathogenesis	References
**SpeB**	**Protease** (cysteine proteinase) Streptopain, streptococcal cysteine proteinase PDB: 4RKX, **4D8B**, 6UKD (w smi), **4D8E** (w smi), 4D8I (w smi), 1DKI (zymogen), 1PVJ (w smi), 6UQD (w smi), 2JTC, 2UZJ Figure 2A	- Chromosomally encoded and highly conserved. - Activates IL-1β. - Cleaves antimicrobial peptides and chemokines. - Activates pyroptosis through gasdermins. - Inhibits autophagy by degrading ubiquitin-binding proteins. - Cleaves antibodies and inhibits effector functions (physiological relevance debated). - Virulence factor in animal models. - SpeB’s contribution to pathogenesis is debated, but its conservation and prevalence indicate an important role.	[15,18,48,86,87,93,94,95,96]
**IdeS/Mac-1**	**Protease** (cysteine proteinase) Immunoglobulin-G-degrading enzyme of *S. pyogenes* PDB: **IY08**, 2AU1, 2AVW Figure 2B	- Chromosomally encoded 35 kDa secreted proteinase. - Prevalent in certain serotypes, including M1, M3, and M49. - Specific hydrolysis of IgG antibodies into Fab’2 and monomeric Fc. - Most likely contributes to immune evasion during infections based on analysis of clinical material, but the lack of activity on some mouse IgG subclasses makes mouse models difficult to interpret.	[19,20,23,25,97,98]
**Mac-2**	**Protease** (cysteine proteinase) **8A47** (IgG Fc) Figure 2B	- Chromosomally encoded allelic variant of Mac-1, chromosomally encoded and prevalent in certain serotypes, including M5, M12, and M28. - Enzymatically impaired variant in some M28 strains. - Probably contributes to immune evasion like IdeS during infection, but not formally tested.	[21,22,99]
**SpyCEP**	**Protease** (serine proteinase) ScpC, CepA PDB: 5XYA, 6VJB (mut), 5XXZ, **5XYR** Figure 4B	- Chromosomally encoded cell envelope 180 kDa proteinase, anchored to the cell wall by sortase. - Cleaves CXCL chemokines with the ELR motif (CXCL1, CXCL2, CXCL6, and CXCL8 (IL-8). - Cleaves the antimicrobial peptide LL-37. - A major vaccine candidate. - Verified virulence factor in mouse models. - Overexpression in hypervirulent isolates.	[6,53,59,61,100,101,102]
**ScpA**	**Protease** (serine proteinase) C5a peptidase PDB: **3EIF**, 7BJ3 (mut), 7YZX (mut), 1XF1 Figure 4A	- Chromosomally encoded conserved 130 kDa cell-wall-anchored proteinase. - Cleaves and inactivates the chemotactic complement factor C5a. - Inhibits chemotactic recruitment of phagocytic cells. - Immunization with ScpA prevents colonization in mice. - Vaccine candidate.	[43,103,104,105]
**EndoS**	**Endoglycosidase** (endo-β-*N*-acetylglucosaminidase, GH18) PDB: 4NUY, 4NUZ (mut), **6EN3** (G2 oligo), 8A64 (IgG Fc), **8A49** (IgG Fc) Figure 3A,B	- Chromosomally encoded 108 kDa endoglycosidase. - Highly conserved (in the majority of serotypes). - Specific hydrolysis of the *N*-linked glycan in the Fc portion of IgG (human, and all mammalian species tested). - Hydrolyzes in the chitobiose core of the IgG-Fc glycan, leaving one *N*-acetylglucosamine and a core fucose (if originally present). Not active on Fab glycans. - Low activity on high-mannose and hybrid-type glycans. - Affects Fc-receptor- and complement-mediated effector functions. - Is a virulence factor in the context of antibody-mediated immunity against bacteria in a mouse model. - Most likely contributes to IgG glycan hydrolysis during severe infections in humans.	[14,15,106,107,108]
**EndoS2**	**Endoglycosidase** (endo-β-*N*-acetylglucosaminidase, GH18) PDB: **6E58**, 6MDS (bia glyc), **6MDV** (hm glyc) Figure 3C,D	- Chromosomally encoded 95 kDa allelic variant of EndoS, conserved in isolates of M49 serotype. - Specific hydrolysis of the *N*-linked glycan in the Fc portion of IgG (human, and all mammalian species tested). Not active on Fab glycans. - Hydrolyzes in the chitobiose core of the IgG glycan, leaving one *N*-acetylglucosamine and a core fucose (if originally present). - Hydrolyzes all glycoforms of IgG, including high-mannose, bisecting, and hybrid-type glycans. - Contribution to virulence not known.	[30,34]
**SodA**	**Superoxide dismutase** No experimental structure.	- Chromosomally encoded. - Intra- and extracellular. - Superoxide resistance.	Not covered in depth here; for a review, see [39]
**AhpC**	**Alkyl hydroperoxidase** No experimental structure.	- Hydrogen peroxide resistance. - Intracellular.	Not covered in depth here; for a review, see [39]
**GpoA**	**Glutathione peroxidase** No experimental structure.	- Superoxide resistance. - Intracellular.	Not covered in depth here; for a review, see [39]
**NoxA**	**NADH oxidase A reductase** No experimental structure.	- Superoxide resistance. - Hydrogen peroxide resistance.	Not covered in depth here; for a review, see [39]
**Sda1**	**Nuclease** Streptococcal nuclease Sda1, Streptodornase PDB: 5FGU, **5FGW** Figure 5	- Phage-encoded in the M1T1 clone. - Cleaves DNA in neutrophil extracellular traps (NETs). - Suppresses TLR9-mediated IFN-α and TNF-α production, leading to decreased macrophage bactericidal activity. - Weakens the recruitment of dendritic cells by reducing IFN-1 levels at the site of infection.	[68,70,109,110,111,112]
**SpnA**	**Nuclease** Streptococcal nuclease A No experimental structure.	- Chromosomally encoded and highly conserved. - Cleaves DNA in neutrophil extracellular traps (NETs). - Animal experiments in mice indicate a role for SpnA in pathogenesis. - Binds IdeS to the streptococcal surface to promote IgG clearance at the bacterial surface.	[67,72,73,113,114,115]
**S5nA**	**Nucleotidase** Streptococcal 5′-nucleotidase A No experimental structure.	- Chromosomally encoded. - Hydrolyzes AMP and ADP to generate the immunomodulatory molecule adenosine. - Acts with SpnA to generate macrophage-toxic deoxyadenosine from DNA. - S5nA’s contribution to pathogenesis is unclear, and it does not seem to have a function in phagocytosis or biofilm formation.	[74,78]
**NADase**	**NAD glycohydrolase** PDB: **4KT6**, **7JI1**, 7WVH (w SLO) Figure 6	- Chromosomally encoded in operon with slo. - Depletes cellular energy storage. - Induces Golgi fragmentation. - Promotes bacterial survival in macrophages. - Translocates into cells together with SLO. - Does not require pore formation by SLO. - High expression in emerging virulent clones. - Human STING (stimulator of interferon genes) genotype affects NADase sensitivity and disease severity.	[82,84,116,117,118,119,120]
**DacA**	**C-di-AMP synthase** No experimental structure.	- Produces C-di-AMP that can be secreted. - Secreted C-di-AMP can activate human STING.	[83,84,121]
**SAGP**	**Arginine deiminase** Streptococcal acid glycoprotein (SAGP) Arginine deiminase (ADI) PDB: **4BOF** Figure 7.	- Inhibits T-lymphocyte differentiation/proliferation. - Promotes bacterial survival at low pH. - Vaccine candidate.	[88,89,91,92,122]

The PDB entries indicated in **bold** are those used for the figures in this review article. Abbreviations: w smi: structure in complex with a small-molecule inhibitor; mut: mutant; IgG Fc: structure in complex with immunoglobulin G (IgG)-fragment crystallizable (Fc) domain; G2 oligo: structure in complex with G2 oligosaccharide; bia glyc: structure in complex with biantennary glycan; hm glyc: structure in complex with high-mannose glycan; w SLO: structure in complex with *Streptococcus pyogenes* Streptolysin O.

## 3. Immunomodulating Enzymes in *S. pyogenes* Pathogenesis

The concept of virulence and virulence factors is a subject of ongoing discussion. A contemporary perspective on microbial pathogenesis focuses on the harm caused to the host, either directly by the pathogen or as a result of the host responses. This so-called damage-response model takes into account both the host immune status and the inherent properties of the pathogen [123,124]. *S. pyogenes* serves as a notable example of a type 3 pathogen within this framework, meaning that both a weak and an overly strong immune response can contribute to host damage and disease severity. Nevertheless, significant harm to the host can also occur during an appropriate immune response, albeit to varying degrees. In light of this perspective, the reductionist approach of searching for individual virulence factors becomes less significant. However, multiple lines of evidence indicate that a number of the immunomodulating enzymes produced by the bacterium play roles in the pathogenesis of *S. pyogenes* infections and contribute to host damage (summarized in Table 1).

As we have described throughout this review, SpeB is the most versatile immunomodulatory enzyme expressed by *S. pyogenes.* Multiple pieces of evidence point to the involvement of SpeB in the pathogenesis of *S. pyogenes* infections. To start, the *speB* gene is universally present and highly conserved and most isolates also produce the enzyme [94,125,126,127,128]. Furthermore, studies measuring seroconversion in patients have confirmed that SpeB is expressed during infections, and a low level of anti-SpeB antibodies appears to be linked with more severe forms of the disease [128,129]. While some research suggests that high levels of SpeB are associated with conditions like STSS, other studies indicate that low SpeB production might correlate with severe infections, possibly by preserving key surface proteins like the M protein [130,131]. It has been observed that SpeB expression decreases when the bacteria are cultured in human blood ex vivo, which could be interpreted as downregulation during systemic infection [132].

Whereas SpeB has ubiquitous roles in the pathogenesis of *S. pyogenes*, the role of the very specific IgG streptococcal cysteine proteinase IdeS has proven more difficult to confirm, even though it is intuitive to assume that IgG hydrolysis has functional consequences. It has been shown that IdeS is active during human severe infections, but it has not been formally established as a virulence factor in animal models, most likely due to the strict specificity for human IgG and the lack of activity on major mouse IgG subclasses [97,98]. Of the other IgG-modifying enzymes, EndoS has been shown to contribute to virulence in a mouse model of infection, but only in the context of partial IgG-mediated immunity against the bacterium [106,108]. Furthermore, during severe human infection, a substantial part of the IgG glycan pool is hydrolyzed, which suggests EndoS involvement in pathogenesis in humans [108]. No study has been conducted concerning the role of EndoS2 in pathogenesis, but, given that its activity is similar to EndoS, it most likely has immunomodulating activity during human infections with M49 *S. pyogenes*.

Concerning enzymes targeting the complement system (excluding SpeB), studies have shown that intranasal vaccination using C5a peptidase can thwart nasopharyngeal colonization by *S. pyogenes* in mice [105]. Additionally, in a long-term colonization mouse model, the absence of the ScpA gene in an *S. pyogenes* strain led to a lower incidence of pneumonia compared to the wild-type bacteria [133].

Focusing on the nucleases, their role in streptococcal pathogenesis is still somewhat unclear. The gene encoding the nuclease SpnA has been identified in all clinical *S. pyogenes* isolates to date [72], and a majority of patients suffering from streptococcal disease develop anti-SpnA antibodies, indicating that SpnA is expressed during infection [67]. Although the exact role of SpnA in the pathogenesis of *S. pyogenes* is still unclear, both blood bactericidal assays and mouse infection models demonstrate that the SpnA knockout strain is less virulent than the parental strain [72]. Whereas both Sda1 and SpnA promote streptococcal survival in whole blood [66,67], the opposite is true for the streptococcal nucleotidase S5nA [78].

Expression of enzymatically active NADase is clearly associated with highly virulent clones of *S. pyogenes* isolated from invasive infections. In line with this, animal infection models have identified the enzymatically active NADase as a significant contributor to virulence [134,135]. Although the detrimental impact of NAD degradation may contribute to inflammation and invasiveness, it is worth noting that NAD is a universally conserved molecule. This suggests that the interaction between NADase and NAD alone is insufficient to explain the variability in disease outcomes seen in patients with invasive *S. pyogenes* infections. Of great interest is, therefore, a recent study showing that NADase expression, in combination with a human STING (stimulator of interferon genes) genotype that is less responsive to c-diAMP, leads to more severe invasive disease [84].

## 4. Immunomodulating Enzymes as Biotechnological Tools

Antibodies are some of the most studied molecules in the life sciences, for their involvement in immunological processes in disease and health, but also for their versatility as tools for studying other biological systems through detection, labeling, localization, targeted delivery, or elimination of unwanted cells or molecules. Therefore, tools to produce, purify, characterize, label, and tailor antibodies are essential biotechnological tools. Since pathogenic bacteria have co-evolved with the human immune system, it is not surprising that numerous such tools originate from bacteria. Prominent examples are the immunoglobulin-binding surface proteins from bacteria such as *Staphylococcus aureus* (Protein A) [136], group G and C streptococci (Protein G) [137], *Finegoldia magna* (Protein L) [138], *Streptococus pyogenes* (Proteins H and Arp) [139,140], and specific IgA proteases (*Neisseria* spp, *Haemophilus influenzae*, *Streptococcus pneumoniae*, and several other bacteria) [141,142] (for reviews, see [11,143]). In addition to these, several of the antibody-active enzymes from *S. pyogenes* described in this review have emerged as very useful biotechnological tools (see Table 2 and Figure 8).

**Table 2 microorganisms-12-00200-t002:** Biotechnological and biopharmacological applications of immunomodulating enzymes.

Enzyme	Application	References
SpeB	- Characterization of human and mouse IgG antibodies.	[144]
IdeS	- Characterization of human IgG antibodies in vitro and in vivo. - Experimental treatment of autoimmunity in animal models. - Treatment of antibody-mediated autoimmune conditions in humans (is an approved drug, as Imlifidase, against kidney transplant rejection). - Vaccine candidate.	[145,146,147,148,149,150,151]
EndoS	- IgG glycan analysis (purified IgG or serum). - Engineering IgG glycosylation by transglycosylase mutants. - Experimental drug against autoimmunity. - Site-specific antibody conjugation through modification of the Fc glycans. - EndoS-fucosidase fusion to defucosylate IgG in vivo.	[150,152,153]
EndoS2	- IgG glycan analysis. - Engineering IgG glycosylation by transglycosylase mutants. - Site-specific antibody conjugation through modification of the Fc glycans.	[30,150,154,155]
SpyCEP	- Vaccine candidate. - Tool to study ELR chemokine function.	[122,156]
ScpA	- Vaccine candidate.	[105,122]
SAGP	- Vaccine candidate.	[122]

**Figure 8 microorganisms-12-00200-f008:**
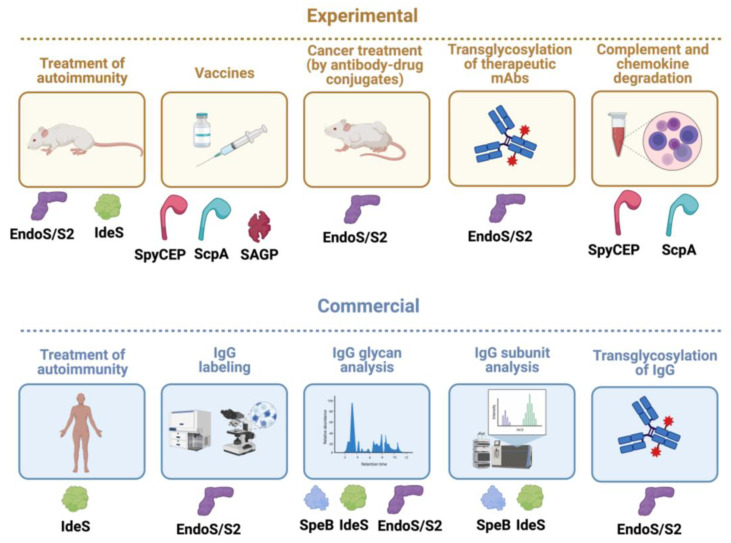
*S. pyogenes* immunomodulating enzymes as biotechnological tools, biological drugs, and vaccines. Given their unique and quite often very specific activity on immunological pathways, several *S. pyogenes* immunomodulating enzymes have been successfully developed as biotechnological tools, biological drugs in themselves, or promising vaccine candidates. Some are still only used experimentally in academic and/or company settings (experimental), while others have been commercialized and are sold as biotechnological products or biological drugs (commercial). The only immunomodulting enzyme that has been approved as a drug against kidney transplant rejection is IdeS, while this enzyme and EndoS have been shown to be efficient in several animal models of antibody-mediated autoimmune diseases. Wild-type EndoS and EndoS2 are widely commercially used to analyze IgG glycans in research settings, and also in developing pipelines for therapeutic monoclonal antibodies (mAbs). The transglycosylation activity of mutated EndoS/EndoS2 has been commercialized for efficient generation of specific glycoforms of IgG (that can have enhanced activity). Several companies are also using this to modify therapeutic mAbs, but none has reached the market as of yet. Commercial products have been launched based on EndoS and EndoS2 together with click chemistry to uniformly label IgG for imaging and FACS/cell sorting. This technique can also be used to generate antibody–drug conjugates (ADC) that could enhance, for instance, mAb-based cancer treatment, but no such ADC has reached the market yet. IdeS and SpeB are widely used products for mass spectrometry and high-pressure liquid chromatography (HPLC) characterization of IgG antibodies from humans and other mammals. The specificity of SpyCEP for chemokines with ELR motifs has been used experimentally to analyze their functions. Analogously, the C5a petidase ScpA can be used to inhibit C5a in experimental systems. Finally, the surface proteins ScpA and SpyCEP and the secreted arginine deiminase SAGP have, alone or in combination, proven to be valuable vaccine candidates, but so far only in animal models of infection. Illustration created using biorender.com.

IdeS has been successfully developed as a biotechnological tool that is extensively used to analyze antibodies for research purposes but also incorporated into many pipelines for the development of therapeutic monoclonal antibodies [144,157].

EndoS and EndoS2 have also been successfully developed for multiple biotechnological uses [158]. EndoS is used extensively for glycan mapping of human and animal IgG, and EndoS2 complements it with a slightly broader specificity for different glycoforms of IgG [30,34,150]. Furthermore, EndoS2 can, in its mutated glycosynthase form, be used to transfer specific glycans to generate homogenous glycoforms of IgG and also to add other payloads (see below) [154,159].

SpeB has proven to be a very useful tool to analyze animal IgG, including from mice, and complements the activity of IdeS, which has limited activity on other-than-human IgG [144].

SpyCEP has been shown to be a very good tool to experimentally modulate the functions of CXCL chemokines with ELR motifs [156]. This cell-wall-anchored enzyme has also emerged as a prime *S. pyogenes* vaccine candidate [101,122].

## 5. Immunomodulating Enzymes as Biological Drugs and Vaccines

Autoimmune diseases pose a significant health burden, impacting around 5% of the global population (as reported by the NIH Autoimmune Coordinating Committee in 2002). Autoimmunity is also a major cause of mortality among young and middle-aged women in developed nations [160]. These conditions are characterized by an immune system that inadvertently targets our own bodies, manifesting in various ways. Many of these diseases are complex, involving both cellular and antibody-based mechanisms for damaging cells and tissues. However, autoantibodies, mainly of the IgG type, contribute to the pathology seen in numerous autoimmune diseases [161]. This is also observed in organ transplant rejection, where antibodies directed against the transplanted organ play a pivotal role [162].

Hence, enzymes like IdeS and EndoS, which specifically break down IgG, present a promising avenue for novel pharmaceuticals targeting IgG-related pathological conditions. Nonetheless, there are several hurdles to overcome before progressing to specific disease models. A crucial consideration is their in vivo effectiveness and specificity to prevent unintended side effects. Early investigations into IdeS and EndoS have demonstrated their high efficiency and excellent tolerance in healthy rabbits, allowing for repeated administration without evident adverse effects [163,164]. Both EndoS and Ides have since been shown, in numerous animal studies of autoimmunity, to, alone or in combination, be very promising biological drugs (Table 2) [163,165,166,167,168,169,170,171,172,173,174,175,176].

IdeS has now been approved as a drug against antibody-mediated transplant rejection, and several clinical trials against other autoimmune conditions are ongoing [145,147,149,177,178] (Figure 8).

The development of a vaccine against *S. pyogenes* is a whole science in itself that cannot be comprehensively covered here (for some starting points, see [101,179,180,181]). It is, however, quite intriguing that several of the immunomodulating enzymes mentioned here, including C5a peptidase, SpyCEP, and SAGP (also called ADI), are being developed alone or in combination as promising vaccine candidates [122,182] (Table 2 and Figure 8). Interestingly, the allelic variant of EndoS in *Streptocoocus equi*, called EndoSe, has shown promise as a vaccine candidate against strangles in horses [183], and the EndoS homologue in *Corynebacterium pseudotuberculosis* CP40 is a promising vaccine candidate against caseous lymphadenitis in goats and sheep [184,185,186,187].

## 6. Concluding Remarks and Future Perspectives

During the process of surveying the literature and writing this review, we have realized that enzymatic immunomodulating activity in *S. pyogenes* is even more complex and multifaceted than we initially anticipated. We have tried to the best of our ability to cover all of the current knowledge (with some stated exceptions) on *S. pyogenes* immunomodulating enzymes with regards to their discovery, modes of action, structure–function relationships, roles in pathogenesis, applications in biotechnology, potential uses as vaccine antigens, and established or potential uses as biological drugs in themselves. However, covering all these aspects exhaustively for all the enzymes is virtually impossible without writing a whole textbook. This piece should therefore be viewed as a starting point for discussing immunomodulating enzymes in *S. pyogenes* and related bacteria. We are hoping that we can stimulate further scrutiny of the literature on the topic, but foremost we are hoping to stimulate researchers to find novel immunomodulating enzymes in pathogenic bacteria. This could, in our minds, contribute to a greater understanding of pathogenic processes during infections and also serve as a source for novel vaccine antigens, biotechnological tools, and innovative biological drugs against immunological disorders.

## Figures and Tables

**Figure 1 microorganisms-12-00200-f001:**
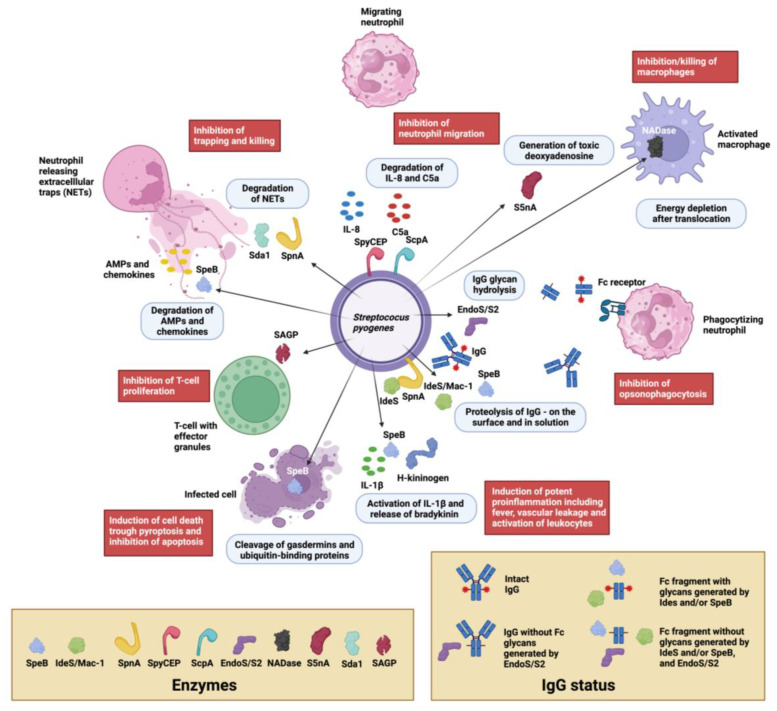
Visual overview of immunomodulating enzymes in *S. pyogenes.* Extracellular enzymes from *S. pyogenes* and their localizations are depicted with cartoons and arrows (enzyme cartoons are defined in the tan box at the lower left). The main enzymatic activities are described in the light blue boxes and the resulting biological outcomes are described in the red boxes. Most enzymatic activities inhibit inflammatory processes, such as neutrophil chemotaxis mediated by IL-8 and C5a (SpyCEP, ScpA), the inhibition of trapping and killing in neutrophil extracellular traps (the DNAases SpnA and SdaI as well as other streptococcal DNAses), the inhibition of killing by antimicrobial peptides (AMPs) and antimicrobial chemokines (both SpeB), the inhibition/killing of macrophages (NADase, S5nA), the inhibition of antibody-mediated killing by neutrophils (opsonophagocytosis) through IgG proteolysis and/or glycan hydrolysis (EndoS/S2, IdeS/Mac-1 and Mac-2, SpeB), SpeB’s hydrolysis of ubiquitin-binding proteins inhibiting apoptosis, and SAGP’s inhibition of T-cell proliferation. Notable exceptions are SpeB’s activity on gasdermins that induce pyroptosis and SpeB’s activity on IL-1β and H-kininogen, which have potent proinflammatory effects. Clarification of IgG status after enzymatic hydrolysis by SpeB, IdeS/Mac-1, and EndoS/S2 are shown in the tan box in the lower right corner. Proteins, cells, and bacteria are not presented to scale. The illustration was created with biorender.com.

**Figure 5 microorganisms-12-00200-f005:**
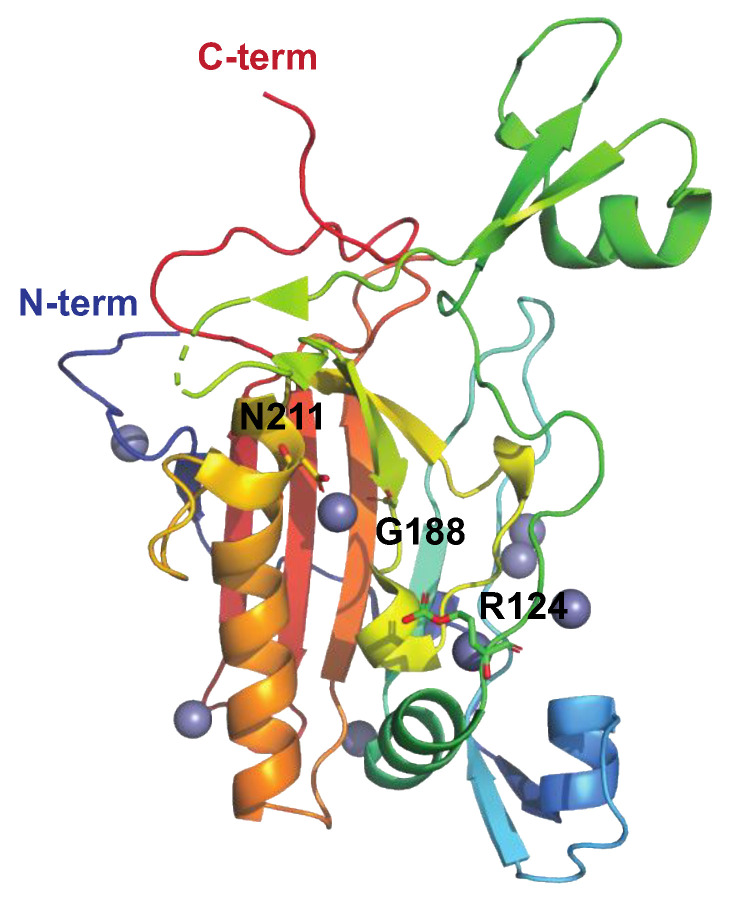
Structure of the streptococcal DNAase Sad1 His188Gly mutant, PDB ID: 5FGW [71], shown in rainbow colors. Sda1-mediated DNA degradation occurs in a sequence-nonspecific manner and is dependent on divalent cations (Zn^2+^ in grey). Asn211, chelating the active site divalent metal cation, is indicated, as is the general base, His188 (here replaced by Gly). Mutating Asn211 generates an inactive enzyme. The putative DNA-binding loop extends from aa 123–137, and Arg124 (side chains shown as sticks) is suggested to stabilize the transition state intermediate during cleavage of the DNA phosphodiester bond.

**Figure 6 microorganisms-12-00200-f006:**
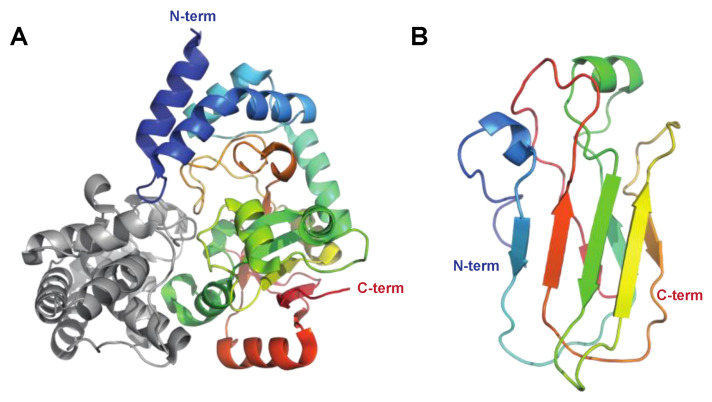
Structure of streptococcal NADase. (**A**) The structure of NADase (shown in rainbow) with the endogenous inhibitor for NADase (IFS, grey) as determined by X-ray crystallography, PDB ID: 4KT6 [81]. Streptococcal NADase consists of two domains, of which the N-terminal domain (aa 1–190) is required for translocation into the host cell, and the C-terminal domain (aa 191–145) harbors the β-NAD^+^ glycohydrolase activity. The crystallized structure only contains the C-terminal catalytic domain. (**B**) The structure for the N-terminal domain as determined by nuclear magnetic resonance (NMR) spectroscopy, PDB ID: 7JI1 [82] shown in rainbow colors.

## Data Availability

No new data were created or analyzed in this study. Data sharing is not applicable to this article.

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
