# Peer review of "Immunomodulating Enzymes from *Streptococcus pyogenes*—In Pathogenesis, as Biotechnological Tools, and as Biological Drugs"

_microorganisms, 2024, doi:10.3390/microorganisms12010200_

Round 1

Reviewer 1 Report

Comments and Suggestions for Authors

The review by Lotta Happonen and Mattias Collin is focused on Streptococcus pyogenes enzymes with direct immunomodulating activities. In general, this review sounds interesting, and addresses many important facets of the complex interaction between the immune response and Streptococcus pyogenes. The work is relevant and of great interest to the field. Such a systematic review in this area will bring important insights and guidance for improving Streptococcus pyogenes associated therapeutics. In my opinion, this manuscript is well written and its contribution is very relevant for the field. In general, I have not major comments on this work. I would only clarify if it is possible to provide more details about literature search, study selection etc. I think that it would help the readers and enhance the quality of review. Finally, I think that the authors should mention about the strength of their review article in the conclusion.

Author Response

We thank the reviewer for the overall positive comments, for the acknowledgment that this review addresses important facts of the complex interaction between the immune response and S. pyogenes, and that the work is deemed as relevant and of great interest to the field bringing important insights and guidance for improving S. pyogenes associated therapeutics. 

We have rewritten some parts of the introduction (lines 46-60), added two new  illustrations (Figures 1 and 8) and added a paragraph on Concluding remarks and Future perspectives. We hope that the reviewer agrees with us that these improvements will collectively help the readers orient themselves in regards to the different enzymes and their functions, enhance the quality of review as well as the highlight the strength of this review.

Reviewer 2 Report

Comments and Suggestions for Authors

This manuscript focuses on immunomodulating enzymes from Streptococcus pyogenes, a human pathogen that causes a wide range of diseases. The review discusses the role of these enzymes in pathogenesis, their potential as biotechnological tools, and their use as biological drugs and vaccines. To improve the manuscript, here are some suggestions:

1. The introduction should be improved by providing clear background why immunomodulating enzymes were focused on and the importance of understanding these enzymes. It should also highlight the gaps in knowledge or research questions that the manuscript aims to address. Additionally, it may be beneficial to provide a brief overview of the content it will be discussed in this manuscript.

2. For 2.1.2, “Immoglobulin glycan hydrolases”, There is a typo. In addition, EndoS and EndoS2 seen to be in the same category with different allelic variants. It could be combined in one paragraph.

3.When discussing the immunoglobulin-degrading and modifying enzymes, it would be better to draw a picture to illustrate the structure of immunoglobulin and the effect of enzymes and their effective site. It would help readers understand the background of antibodies and the function of bacterial enzymes.

4. For each enzyme, the authors mainly discuss the structure and effect of these enzymes. It would be better to integrate the paragraphs on pathogenesis, its implications in biotechnological tools, and biological drugs and vaccines of each enzyme into their category for better understanding the effect and function of the enzymes

5. In the section of “2.3 Enzymes acting on chromatin and cellular processes,” it would be better to provide a brief background about chromatin and cellular processes with immune modulation, then describe the function of these enzymes.

6. For better understanding, a clear picture of these enzymes with immunomodulating activity should be drawn to illustrate their location, function, and potential implications.

7. In the section of 2.3.3.1 SpeB activity on gasdermins, SpeB activates gasdermins, which have a positive role in the active immune system, which has the opposite role of the antibody cleavage. These should be discussed for their role in pathogenesis.

8. Provide some perspective for future directions, like unanswered questions, potential therapeutic applications, or the development of novel therapeutic strategies to target these enzymes.

9. Full names should be provided for these enzymes.

Author Response

We thank the reviewer for the insightful comments. 

1. The introduction should be improved by providing clear background why immunomodulating enzymes were focused on and the importance of understanding these enzymes. It should also highlight the gaps in knowledge or research questions that the manuscript aims to address. Additionally, it may be beneficial to provide a brief overview of the content it will be discussed in this manuscript.

We thank the review for the comment. We present the rationale of writing this review article in the submitted manuscript on lines 46-55 in the Introduction. These statements and objectives have been emphasised on lines 46-60 in the resubmission. In this text passage, we have also added a brief overview of the content discussed.

2. For 2.1.2, “Immoglobulin glycan hydrolases”, There is a typo. In addition, EndoS and EndoS2seen to be in the same category with different allelic variants. It could be combined in one paragraph.

We thank the reviewer for the comment. We have fixed the typo. We do, however, argue that the paragraphs on EndoS and EndoS2 should be kept separate, as even though they are allelic variants. EndoS2 is only approximately 40% identical to EndoS on the amino acid level. Furthermore, EndoS2 has a much broader specificity for IgG-Fc glycans than EndoS by hydrolyzing bulkier glycans such as high mannose and bisecting structures. 

3. When discussing the immunoglobulin-degrading and modifying enzymes, it would be better to draw a picture to illustrate the structure of immunoglobulin and the effect of enzymes and their effective site. It would help readers understand the background of antibodies and the function of bacterial enzymes.

We thank the reviewer for this comment. We have now added a new illustration (Figure 1) to highlight the structure of immunoglobulins and the effects of the different enzymes (IdeS, SpeB, EndoS, EndoS2) and their effective sites on these. 

4. For each enzyme, the authors mainly discuss the structure and effect of these enzymes. It would be better to integrate the paragraphs on pathogenesis, its implications in biotechnological tools, and biological drugs and vaccines of each enzyme into their category for better understanding the effect and function of the enzymes.

During the writing process of this review article, different layouts of the text were tested, where one option was to describe each enzyme individually on all different aspects in succession (pathogenesis, its implications as a biotechnological tool and as a biological drug or vaccine). This layout did, however, produce a less reader friendly version of the text, and read more like a catalogue than a descriptive overview of the field. To summarise our main ideas, we generated Tables 1 and 2 as an overview (present in the original submission). 

5. In the section of “2.3 Enzymes acting on chromatin and cellular processes,” it would be better to provide a brief background about chromatin and cellular processes with immune modulation, then describe the function of these enzymes.

We thank the reviewer for this comment. This text paragraph exists in the original submission on lines 300-314, but has probably been missed due to us placing the subheading for 2.3.1 Enzymes acting on DNA and nucleotides in the wrong place. This has now been fixed. 

6. For better understanding, a clear picture of these enzymes with immunomodulating activity should be drawn to illustrate their location, function, and potential implications.

We thank the reviewer for this comment. We have now added an overview figure at the beginning of the manuscript to illustrate the location, function, and potential implications of the different immunomodulating enzymes discussed (new Figure 1).

7. In the section of 2.3.3.1 SpeB activity on gasdermins, SpeB activates gasdermins, which have a positive role in the active immune system, which has the opposite role of the antibody cleavage. These should be discussed for their role in pathogenesis.

We agree that the potential host beneficial role for SpeB activity on gasdermins should be highlighted. In addition to the mention in the main text we have now also highlighted this in the overview Figure 1.

8. Provide some perspective for future directions, like unanswered questions, potential therapeutic applications, or the development of novel therapeutic strategies to target these enzymes.

Great suggestion! We have added a concluding section discussion the limitations and future development we are hoping for.

9. Full names should be provided for these enzymes.

Full names for each enzyme is given in Table 1. The Mac-1 and Mac-2 proteins are called Mac based on the corresponding gene (mac), as is EndoS (ndoS) and EndoS2 (ndoS2).     

Reviewer 3 Report

Comments and Suggestions for Authors

The detailed review by Happonen and Collin on immunomodulating enzymes from Streptococcus pyogenes is very comprehensive and greatly helps in understanding the field. Therefore, it is well suited for publication in Microorganisms, but the following revision is requested.

Each enzyme is listed (Tables) and their structures are depicted (Figures), but there is no illustration of the enzyme's function. For example, a simple illustration of SpeB degrading Ig and IgG would help readers better understand the enzyme and make this a more widely cited review. Additional illustrations are requested throughout the paper, including biotechnological tools and drug applications.

Author Response

We thank the reviewer for the positive comments on the manuscript.  

To increase the readability of the paper, we have now included two new illustrations, Figure 1 (overview figure of enzyme location and function as well as IgG degradation/modification) and Figure 8 (figure of enzymes as biotechnological tools, biological drugs and vaccines) in this resubmission.

Round 2

Reviewer 2 Report

Comments and Suggestions for Authors

The authors have addressed my concerns.